# Clinical and pharmacological factors associated with mortality in patients with COVID-19 in a high complexity hospital in Manaus: A retrospective study

Rebeka Caribé Badin[1]* , Robson Luís Oliveira de Amorim[1], Alian Aguila[2], Liliane Rosa Alves Manaças[3]

1 Department of Neurosurgery, Getúlio Vargas University Hospital, Manaus, Amazonas, Brazil,
2 Department of Cardiology, Memorial Hospital System, Hollywood, Florida, United States of America,
3 Department of Pharmacology, Brazilian National Cancer Institute José Alencar Gomes da Silva (INCA)/ Hospital II, Rio de Janeiro, Rio de Janeiro, Brazil

These authors contributed equally to this work.
* rebekaaalves@hotmail.com

**Data Availability Statement:** All relevant data are within the paper and its Supporting Information files.

## Abstract

COVID-19 is a contagious infection caused by the SARS-CoV-2 virus, responsible for more than 5 million deaths worldwide, and has been a significant challenge for healthcare systems worldwide. Characterized by multiple manifestations, the most common symptoms are fever, cough, anosmia, ageusia, and myalgia. However, several organs can be affected in more severe cases, causing encephalitis, myocarditis, respiratory distress, hypercoagulable state, pulmonary embolism, and stroke. Despite efforts to identify appropriate clinical protocols for its management, there are still no fully effective therapies to prevent patient death. The objective of this study was to describe the demographic, clinical, and pharmacotherapeutic management characteristics employed in patients hospitalized for diagnosis of COVID-19, in addition to identifying predictive factors for mortality. This is a single-center, retrospective cohort study carried out in a reference hospital belonging to the Brazilian public health system, in Manaus, from March 2020 to July 2021. Data were obtained from analyzing medical records, physical and electronic forms, medical prescriptions, and antimicrobial use authorization forms. During the study period, 530 patients were included, 51.70% male, with a mean age of 58.74 ± 15.91 years. The overall mortality rate was 23.58%. The variables age, number of comorbidities, admission to the ICU, length of stay, oxygen saturation, serum aspartate transaminase, and use of mechanical ventilation showed a positive correlation with the mortality rate. Regarding pharmacological management, 88.49% of patients used corticosteroids, 86.79% used antimicrobials, 94.15% used anticoagulant therapy, and 3.77% used immunotherapy. Interestingly, two specific classes of antibiotics showed a positive correlation with the mortality rate: penicillins and glycopeptides. After multivariate logistic regression analysis, age, number of comorbidities, need for mechanical ventilation, length of hospital stay, and penicillin or glycopeptide antibiotics use were associated with mortality (AUC = 0.958).

**Funding:** The author(s) received no specific funding for this work.

## Introduction

COVID-19 disease caused by severe acute respiratory syndrome coronavirus 2 (SARS-CoV-2 —Severe Acute Respiratory Syndrome Coronavirus 2) has been a threat to global health [1–4], having its beginning in Wuhan, China, and quickly spreading around the world [5]. The clinical picture triggered by this virus is highly heterogeneous, ranging from asymptomatic infection to life-threatening complications [6]. According to the WHO, until the end of 2021, 279 million cases of COVID-19 and about 5.4 million deaths were confirmed worldwide. In Brazil, we had 22.2 million cases in this same period and about 618 thousand deaths, of which 13.8 thousand occurred in the state of Amazonas [7]. COVID-19 is classified as mild, moderate, severe, and critical according to clinical, radiological, and laboratory findings. Fortunately, many of the infected population have a mild condition [8]. The most common symptoms are fever, cough, anosmia, ageusia, and myalgia. However, several organs can be affected in more severe cases, resulting in encephalitis, myocarditis, respiratory distress, and a hypercoagulable state. The latter can cause pulmonary embolism and stroke [8, 9]. The exacerbated inflammatory response, characteristic of SARS-CoV-2 infection, is due to the increase in pro-inflammatory cytokines and chemokines (IL-1, IL-6, IFN-γ, and TNF-α) that can lead to severe acute respiratory syndrome (SARS) [10]. Regarding pharmacotherapeutic management, the principal used treatment options were corticosteroids, antimalarials, anticoagulants, immunomodulators, vitamins C and D, antimicrobials, interleukin-6 inhibitors, and convalescent plasma therapy [11, 12]. Several studies reported the association between comorbidities (hypertension, type 2 diabetes, cardiovascular disease, and obesity) and age over 65 years to disease severity [13–19].

However, in high-income countries, the mortality rate for COVID-19 has reduced over time, a trend not followed in lower-income countries [1, 20]. Therefore, the objective of this study was to identify predictive factors for mortality in patients hospitalized with COVID-19 in a Brazilian hospital of high complexity, located in the city of Manaus, one of the epicenters of South America, as well as to describe demographic, clinical, and pharmacotherapeutic management characteristics used in these patients.

## Materials and methods

### Design study and population

In this retrospective cohort, we consulted the electronic medical records of patients hospitalized with COVID-19 infection between March 2020 and July 2021. The extended period of the survey intended to include the two exponential waves of COVID-19 cases in the city of Manaus. The study was carried out at Getúlio Vargas University Hospital, which belongs to the Unified Health System and is managed by the Brazilian Hospital Services Company. The hospital is a reference both in medium and high complexity care and in the training and qualification of health professionals. Its ICU capacity is 10 adult beds—TYPE II, but it was later enabled for 30 adult ICU II beds—severe acute respiratory syndrome (SARS), COVID-19, according to the National Registry of Health Establishments (CNES). The research was approved by the Research Ethics Committee of the Federal University of Amazonas, under protocol number CAAE n° 57750422.4.0000.5020.

### Source data and outcomes

Data collection included medical records, prescriptions, and antimicrobial requests sent to the Hospital Infection Control Service (SCIH) and the electronic system AGHU (Application for Management of University Hospitals). For each patient, we gathered the following information:

age, sex, comorbidities, date of admission to the intensive care unit, mechanical ventilation support, length of stay, use of antimicrobials, corticosteroids, anticoagulants, immunomodulators, blood oxygen saturation, aspartate aminotransferase (AST), and alanine aminotransferase (ALT) serum levels. We chose hospital mortality as the primary endpoint. The length of stay was set from the date of admission to hospital discharge, transfer to other hospitals, or death.

## Statistical analysis

We calculated the mean and standard deviation for the quantitative variables, and the Kolmogorov-Smirnov Normality test was applied. Student's t-test (Normal Distribution) and Mann-Whitney (Non-Normal Distribution) were used to compare two groups (survivors and non-survivors). Categorical variables were shown in percentages or absolute values, and the Chi-Square Test and Fisher's Exact Test were used to verify the association between categorical variables. We employed the logistic regression model and calculated the receiver operating characteristic (ROC) curve. The pseudo R2 was calculated to verify the model's explanatory power. All p values < 0.05 were considered statistically significant. For the analyses, we used the statistical program Stata® version 17.

## Results

### Clinical and demographic cohort characteristics

In total, five hundred and thirty patients were included in the study; there was a slight predominance of males (51.70%), the mean age of the patients was 58.74 ± 15.91 years, and about 52% were 60 years old or older. The overall mortality rate was 23.58%, and most were men (52.00%); for patients over 60 years of age, the mortality rate was 76.80%. The mean overall length of stay was 14.83 ± 14.76 days, ranging from 13.90 ± 14.77 days and 17.82 ± 14.37 for the survivors and non-survivors, respectively.

Regarding comorbidities, 64.91% of patients had some comorbidity, 61.98% and 74.40% for the survivors and non-survivors groups, respectively. Most patients had one to two comorbidities (52.70%), with hypertension (46.04%) and diabetes (29.06%) being the most prevalent in the study. The presence of hypertension for the survivors was 44.44%, and for the non-survivors, it was 51.20%. While the presence of diabetes was 28.15% and 32.00% in the group of survivors and non-survivors, respectively.

About 35% of patients required intensive care, with 18.02% and 88.80% in the survivors and non-survivors groups, respectively. Of the patients admitted to the Intensive Care Unit (ICU), approximately 76% required mechanical ventilation. The median of oxygen saturation on the admission was 95%, varying between 96% and 94% in the survivors and non-survivors groups, respectively. Concerning serum AST and ALT levels, the median in survivors was 34 U/L and 49 U/L; and the non-survivors group was 43.5 U/L and 45 U/L, respectively. Statistically significant associations were observed between the mortality rate and the variables age (p<0.001), length of hospital stay (p<0.001), presence of comorbidity (p = 0.011), number of comorbidities (p<0.001), need for admission on ICU (p<0.001), mechanical ventilation (p<0.001), oxygen saturation (p<0.001) or AST (p = 0.001). However, there was no such association regarding sex (p = 0.938). Regarding comorbidities, there was no association between mortality and hypertension (p = 0.185), diabetes (0.407), or obesity (p = 0.086) (Table 1).

### COVID-19 pharmacological management

About the pharmacotherapeutic management used for COVID-19, about 88% of the patients in the study used corticosteroids, 86.91% and 93.60% in the survivors and non-

**Table 1. Demographic and clinical characteristics of patients diagnosed with COVID-19.**

| Variables | Overall | Survivor | Non-survivor | p-value |
|---|---|---|---|---|
| **Number of patients** | 530 | 405 | 125 | |
| **Age, mean (SD)** | 58.74 ± 15.91 | 56.25 ± 15.64 | 66.81 ± 14.05 | **< 0.001** |
| **Age, (%)** | | | | **< 0.001** |
| < 60 | 47.74 | 55.31 | 23.20 | |
| ≥ 60 | 52.26 | 44.69 | 76.80 | |
| **Sex (%)** | | | | 0.938 |
| Men | 51.70 | 51.60 | 52.00 | |
| Woman | 48.30 | 48.40 | 48.00 | |
| **Comorbidities (%)** | | | | **0.011** |
| Yes | 64.91 | 61.98 | 74.40 | |
| No | 35.09 | 38.02 | 25.60 | |
| **Number of comorbidities (%)** | | | | **< 0.001** |
| 0 | 35.09 | 38.02 | 25.60 | |
| 1 | 27.92 | 29.63 | 22.40 | |
| 2 | 24.72 | 23.70 | 28.00 | |
| 3 | 9.06 | 7.41 | 14.40 | |
| 4 | 3.02 | 1.23 | 8.80 | |
| > 4 | 0.19 | 0 | 0.80 | |
| **Comorbidities (%)** | | | | |
| HAS | 46.04 | 44.44 | 51.20 | 0.185 |
| DM | 29.06 | 28.15 | 32.00 | 0.407 |
| Obesity | 6.98 | 5.93 | 10.40 | 0.086 |
| HAS +DM | 20.38 | 20.74 | 19.20 | 0. 709 |
| DM + Obesity | 0.38 | 0.25 | 0.80 | 0.416 |
| HAS +DM + Obesity | 2.08 | 1.73 | 3.2 | 0.297 |
| **ICU admission (%)** | | | | **< 0.001** |
| Yes | 34.72 | 18.02 | 88.80 | |
| No | 65.28 | 81.98 | 11.20 | |
| **Mechanism ventilation (%)** | | | | **< 0.001** |
| Yes | 26.42 | 7.90 | 86.40 | |
| No | 73.58 | 92.10 | 13.60 | |

HAS: sistemic arterial hipertension, DM: Diabetes Melitus.

survivors groups, respectively. There was a statistically significant association between corticosteroids and mortality (p = 0.041). Antimicrobials were used in 86.79% of the patients, varying between 83.95% and 96.00% in the survivors and non-survivors groups. The number of antimicrobials used varied between 1 and 8. Most patients (69.24%) used 1 to 3 antimicrobials, 10.75% used 4 to 5 antimicrobials, and 6.79% used more than five antimicrobials. In the survivor group, 75.80% of patients used 1 to 3 antimicrobials, 5.68% used 4 to 5, and 2.47% used more than five antimicrobials. Among the non-survivors, 48.00% used 1 to 3 antimicrobials, 27.20% used 4 to 5 antimicrobials, and 20.80% used more than five antimicrobials. Regarding the use of antibiotics, the most used classes were cephalosporins (70.00%), followed by macrolides (54.72%), penicillins (33.40%), and glycopeptides (15.47%). This usage profile was slightly different in the non-survivor group. It is worth noting that, in many cases, the patient had antibiotic combinations or sequential antibiotic therapy regimens due to complications arising from the COVID-19 infection. A

**Table 2. Pharmacotherapeutic classes used in the treatment of COVID-19 infection.**

| Variables | Overall | Survivors | Non-survivors | p-value |
|---|---|---|---|---|
| **Corticosteroids** | | | | **0.041** |
| Yes | 88.49 | 86.91 | 93.60 | |
| No | 11.51 | 13.09 | 6.40 | |
| **Antimicrobials** | | | | **0.001** |
| Yes | 86.79 | 83.95 | 96.00 | |
| No | 13.21 | 16.05 | 4.00 | |
| **Number of Antimicrobials** | | | | **< 0.001** |
| 0 | 13.21 | 16.05 | 4.00 | |
| 1 | 16.23 | 18.02 | 10.40 | |
| 2–3 | 53.01 | 57.78 | 37.60 | |
| 4–5 | 10.75 | 5.68 | 27.20 | |
| > 5 | 6.79 | 2.47 | 20.80 | |
| **Classes of Antibiotics** | | | | |
| Penicillins | 33.40 | 19.01 | 80.00 | **< 0.001** |
| Cephalosporins | 70.00 | 69.14 | 72.80 | 0.435 |
| Macrolides | 54.72 | 54.57 | 55.20 | 0.901 |
| Glycopeptides | 15.47 | 5.93 | 46.40 | **< 0.001** |
| **Antifungal** | | | | **< 0.001** |
| Yes | 8.68 | 5.93 | 17.60 | |
| No | 91.32 | 94.07 | 82.40 | |
| **Therapeutic anticoagulation** | | | | **0.021** |
| Yes | 94.15 | 92.84 | 98.40 | |
| No | 5.85 | 7.16 | 1.60 | |
| **Therapeutic immunomodulator** | | | | 0.472 |
| Yes | 3.77 | 3.95 | 3.20 | |
| No | 96.23 | 96.05 | 96.80 | |

small percentage of patients used antifungal therapy (8.68%). There was a statistically significant association between antimicrobial use and mortality ($p < 0.001$). When we consider the antibiotic classes individually, there was no association between the use of cephalosporins ($p = 0.435$) and macrolides ($p = 0.901$) with mortality; on the other hand, there was a statistically significant association between the use of the glycopeptide class ($p < 0.001$) and penicillins ($p < 0.001$) with mortality. There was also a significant association between antifungal use and mortality ($p < 0.001$).

Anticoagulant therapy was used in 94.15% of the patients, while immunomodulatory therapy was used rarely (3.77%). There was a statistically significant association between the use of anticoagulants and mortality ($p = 0.021$); however, there was no association between mortality and the use of immunomodulators ($p = 0.472$) (Table 2).

## Characteristics associated with mortality

The statistically significant variables identified in the univariate analysis were included in the multivariate logistic regression analysis (model 1) (S1 Table). However, some variables did not prove to be statistically significant in model one, such as the need for ICU, use of corticosteroids, antimicrobials, anticoagulants, antifungals, and antibiotic class of glycopeptides. Thus, a second multivariate logistic regression analysis model was done using the reverse stepwise approach until all variables were significant (Table 3).

**Table 3. Multivariate logistic regression analysis of risk factors for mortality in hospitalized patients with COVID-19.**

| Variable | OR | SE | p-value | Confidence Interval (95%) | |
|---|---|---|---|---|---|
| | | | | Lower | Upper |
| Age | 1.059 | 0.0142 | < 0.001 | 1.031 | 1.087 |
| Number of comorbidities | 1.497 | 0.249 | 0.015 | 1.080 | 2.075 |
| Length of stay | 0.923 | 0.0148 | < 0.001 | 0.894 | 0.952 |
| VM | 79.709 | 39.128 | < 0.001 | 30.455 | 208.615 |
| Penicillin | 4.53 | 2.039 | 0.001 | 1.872 | 10.943 |
| Glycopeptides | 3.257 | 1.722 | 0.026 | 1.155 | 9.182 |

SE: standard error MV: mechanical ventilation

The pseudo R2 of the multivariate logistic regression model 2 was 0.629. The area under curve was 0.958, the sensitivity and specificity were 83.20% and 95.80%, respectively (Fig 1).

## Discussion

In this single-center study carried out in a high-complexity hospital, we analyzed the behavior of COVID-19 in 530 patients who required hospitalization. The overall mortality rate was 23.58%, and for patients who required intensive care, it was 60.33%. In a systematic review covering China, France, Denmark, the Netherlands, Hong Kong, Italy, Singapore, Spain, and

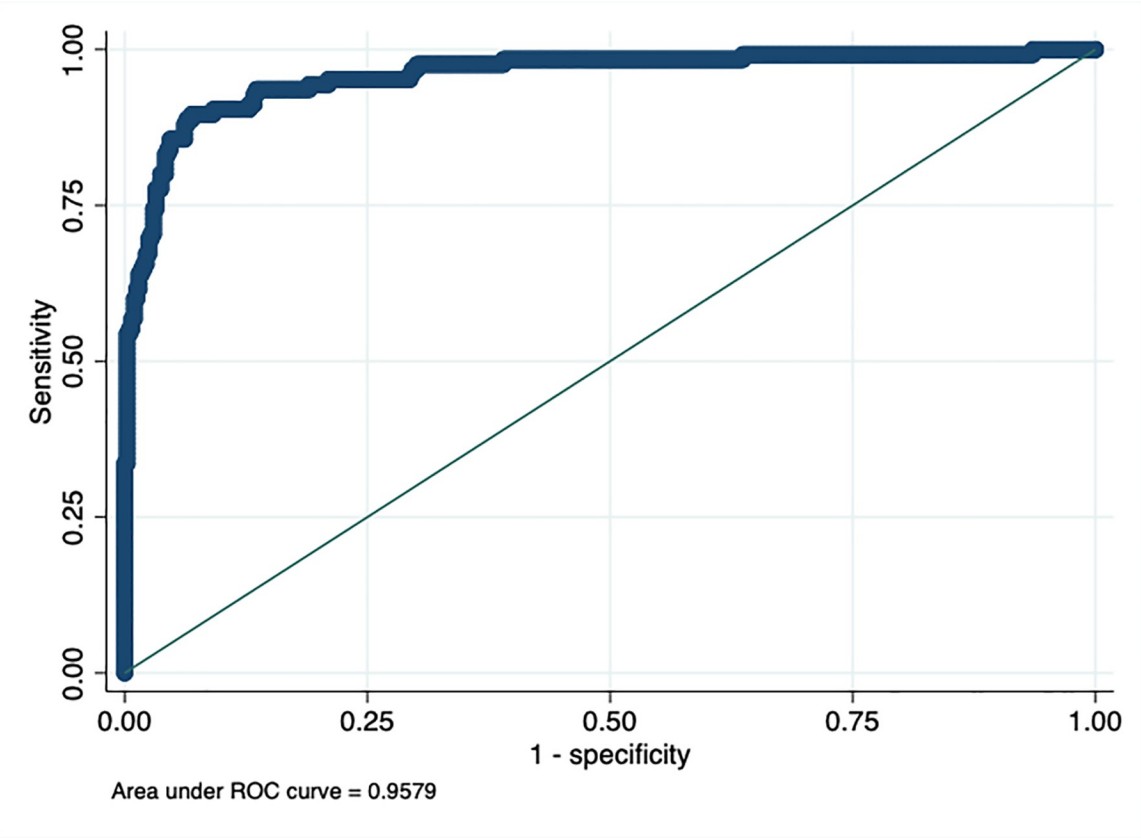

**Fig 1. Receiver operating curve (ROC) multivariate logistic regression model of risk factors.**

the United Kingdom, mortality rates ranged from 0 to 84.6% in the intensive care unit [21], indicating a high diversity in the global mortality rate for patients with COVID admitted to the ICU [22]. Our study covered an extended period, including the beginning of the pandemic marked by the absence of vaccination and without scientific evidence of the most appropriate treatment. According to epidemiological data, the behavior of the COVID-19 epidemic in Manaus was characterized by two exponential growth curves of cases. The first wave occurred between March and May 2020, and the most prevalent variant was B.1,195, followed by a decline and stability in the number of cases between June and November 2020, and a second wave between December 2020 and February 2021, with a predominance of the P.1 variant [23–25].

There was a high prevalence of mortality in elderly patients ($\geq$ 60 years) (76.80%), as was also observed in other studies [1, 9, 26–28].

With advancing age, humoral and cellular immune functions decrease, and there is a decrease in T cells, immunoglobulin M and interferon levels, leading to a greater risk of death [29]. Immunosenescence impairs the recognition and elimination of pathogens; in addition, there is a chronic increase in systemic inflammation called inflammatory aging, possibly contributing to the severity of COVID-19 [1]. In the logistic regression analysis, age obtained an odds ratio of 1.059 (95% CI: 1,031–1,087).

Regarding gender, we observed a slight predominance of males, but mortality rates did not differ between genders. In the systematic review by Tan et al. [22], which included 45 studies with 16,561 ICU patients in 17 countries, it was possible to identify that more than two-thirds of the patients were male. Still, mortality rates did not differ between the sexes, in line with the present study. In contrast, several studies report higher mortality for men [9, 30–32], and the study by Baguma et al. [1] reported higher mortality for women, suggesting that the impact of sex on mortality depends on other factors characteristic of the analyzed population.

The mean length of stay was statistically different between survivors and non-survivors; patients who stayed longer in the hospital had a better clinical outcome, similar to the results of other studies [33, 34], indicating that the ability to provide hospital care for patients is a priority to prevent inequalities in mortality rates [22]. In logistic regression analysis, length of stay was a predictor of favorable clinical outcome (OR 0.923, 95% CI: 0.894–0.952).

Most patients had comorbidities (64.91%), and the presence and number of comorbidities were statistically significant for mortality, similar to what has been reported in other studies [1, 22, 28]. In addition, the number of comorbidities was statistically significant in the multiple logistic regression (OR 1.497, 95% CI: 1.080–2.075).

Despite the high prevalence of comorbidities hypertension (46.04%) and diabetes (29.06%), both did not show a statistically significant association with the outcome of death, contrary to the results of other studies [1, 22, 35]. However, there are gaps to be filled in the impact of hypertension on the prognosis for COVID-19 [36].

About 35% of the study patients required intensive care, 18.02% and 88.80% in the survivor and non-survivor groups, respectively. Of the patients admitted to the ICU, approximately 76% required mechanical ventilation; the need for admission to the ICU and ventilation were negative predictors of clinical outcomes. Following the same trend, patients who required mechanical ventilation had a worse prognosis in other studies [37–39]. Multiple factors influence ICU mortality, ranging from individual susceptibility to care team training. There are reports of almost 100% mortality in patients who required mechanical ventilation in studies during the global pandemic peak of the disease [33, 40, 41]. One of the possible reasons for these findings may be the lack of experience and inadequate treatment of SARS related to COVID-19, combined with late admission to the ICU [33]. In the multiple logistic regression,

the need for intensive care was not statistically significant, while mechanical ventilation was statistically significant (OR 79.709, 95% CI: 30.455–208.615).

Oxygen saturation and aspartate aminotransferase were statistically significant for mortality, similar to findings from other studies (1, 37). However, in the multiple logistic regression, these parameters were not statistically significant.

Regarding the pharmacotherapeutic treatment used in these patients, about 88% used corticosteroids; there was a significant association between the use of this class of medication and mortality. In the CODEX study that analyzed the use of dexamethasone in patients with COVID-19, the mortality rate was high, and there was no significant difference between the groups that used or did not use dexamethasone. However, it resulted in increased survival and days free of mechanical ventilation [42].

In the RECOVERY study, dexamethasone generated a marked reduction in mortality in patients who required mechanical ventilation but not among those who did not receive respiratory support [43].

In the systematic review by Crichton et al. [44], it was possible to identify that the use of corticosteroids (dexamethasone, methylprednisolone, hydrocortisone) had a beneficial effect on mortality, especially in patients who required mechanical ventilation. However, the relevance of this benefit may not be the same in all environments [22]. An important aspect is the dose administered; in a study on patients with non-COVID viral pneumonia, high doses of dexamethasone were associated with increased mortality [45]. Therefore, the present study did not assess the administered dose of corticosteroids only if there was this pharmacological approach, which may have influenced the result.

The general use of antimicrobials occurred in 86.8% of the patients; the main classes of antibiotics were cephalosporins, macrolides, penicillins, and glycopeptides. A systematic review and meta-analysis performed by Langford and colleagues in 2020 estimated an 8.6% risk of bacterial and fungal co-infection in patients with COVID-19 [46]. Other studies have also reported this low incidence [47–49], differentiating from influenza, which has a high prevalence of co-infection [50]. However, many hospitalized patients with COVID-19 received antimicrobial therapy; it is estimated that virtually everyone in the intensive care unit was exposed to this therapy [46, 51]. Furthermore, a substantial increase in the use of antimicrobials in the hospital environment was detected in other studies carried out in Portugal [52], England [53], Spain [50] and Pakistan [2]. The difference between sepsis' signs and symptoms and the worsening of COVID-19 is not always easy; for this reason, the empirical administration of antimicrobials occurs in many patients with COVID-19 [2]. In the present study, there was a statistically significant association between the use of antimicrobials and mortality. In addition, the analysis of antibiotic classes separately revealed an association between the use of penicillins or glycopeptides and the negative outcome of death. At the moment, few studies have assessed the impact of antibiotic use in patients with COVID-19; among them, most evaluate the use of azithromycin, a macrolide widely cited in the literature as a therapeutic approach for SARS-CoV-2 infections [54]. It is unknown whether this result is related to adverse effects related to the use of antimicrobials or whether it is due to the worsening of the disease. However, there are reports that the use of antibiotics without bacterial infection can lead to a cytokine storm and that early antibiotic use in patients with non-severe COVID-19 is associated with a risk for disease progression [54–56].

The SARS-CoV-2 infection is associated with thromboembolic complications affecting various organs such as the lungs, heart, and brain. Literature data indicate that the risk of hospitalized patients developing venous thromboembolism varies from 7.9% to 22.7%, with the highest prevalence associated with ICU admission. Drug prophylaxis includes using unfractionated heparin or low-molecular-weight heparin [57, 58]. Anticoagulant therapy was used in

94.2% of the study patients, ranging from 92.8% to 98.4% in the survivors and non-survivors groups. There was a significant association between anticoagulant use and mortality, in line with the findings by Musoke et al. [59]. Several studies report that the use of anticoagulant therapy reduces the mortality of patients with severe COVID-19 [3, 60–62], while the study by Tremblay et al. [63] does not report these benefits. These divergences in the effects of anticoagulant therapy may depend on the balance of identifying patients at greater risk of hemorrhage or greater chance of thrombosis [59]. Additionally, there is no consensus on whether using anticoagulants above the standard prophylactic dose reduces mortality [59, 64, 65].

The main limitation of this study was the lack of evaluation of the administered doses of corticosteroids, anticoagulants, and antimicrobials. Furthermore, the study was unicentric and retrospective. It would be interesting for future work to explore the relationship between different genotypes of SARS-CoV-2, laboratory and microbiological parameters with mortality rate.

According to the findings of this study, we conclude that, in the face of infection with highly varied signs and symptoms and directly influenced by multifactorial components related to viral characteristics and the immune response of patients, despite the constant search for therapeutic approaches based on clinical evidence, we still have several points to be clarified. Our data reaffirm the need for greater attention to elderly patients and those with comorbidities, revealing the need for individualizing pharmacotherapeutic management, especially for anticoagulants, corticosteroids, and antimicrobials. The correlation between the use of penicillins or glycopeptides antibiotics and a higher mortality rate was an essential contribution of this work. In addition, our data corroborate the findings that the need for mechanical ventilation indicates a worse prognosis, while a more extended hospital stay is associated with a favorable clinical outcome. Studies that contribute to a more assertive clinical approach in the treatment of COVID-19 are essential to promote a reduction in mortality associated with this virus.

## Supporting information

**S1 Table. Multiple logistic regression analysis of risk factors for mortality in hospitalized patients with COVID-19.**
(DOCX)

## Acknowledgments

Multidisciplinary health team at Getúlio Vargas University Hospital.

## Author Contributions

**Conceptualization:** Rebeka Caribé Badin, Robson Luís Oliveira de Amorim, Alian Aguila, Liliane Rosa Alves Manaças.

**Data curation:** Rebeka Caribé Badin, Robson Luís Oliveira de Amorim, Alian Aguila, Liliane Rosa Alves Manaças.

**Formal analysis:** Rebeka Caribé Badin, Robson Luís Oliveira de Amorim, Alian Aguila, Liliane Rosa Alves Manaças.

**Investigation:** Rebeka Caribé Badin, Robson Luís Oliveira de Amorim, Alian Aguila, Liliane Rosa Alves Manaças.

**Methodology:** Rebeka Caribé Badin, Robson Luís Oliveira de Amorim, Alian Aguila, Liliane Rosa Alves Manaças.

**Project administration:** Rebeka Caribé Badin, Robson Luís Oliveira de Amorim, Alian Aguila, Liliane Rosa Alves Manaças.

**Resources:** Rebeka Caribé Badin, Robson Luís Oliveira de Amorim, Alian Aguila, Liliane Rosa Alves Manaças.

**Supervision:** Rebeka Caribé Badin, Robson Luís Oliveira de Amorim, Alian Aguila, Liliane Rosa Alves Manaças.

**Validation:** Rebeka Caribé Badin, Robson Luís Oliveira de Amorim, Alian Aguila, Liliane Rosa Alves Manaças.

**Visualization:** Rebeka Caribé Badin, Robson Luís Oliveira de Amorim, Alian Aguila, Liliane Rosa Alves Manaças.

**Writing – original draft:** Rebeka Caribé Badin, Robson Luís Oliveira de Amorim, Alian Aguila, Liliane Rosa Alves Manaças.

**Writing – review & editing:** Rebeka Caribé Badin, Robson Luís Oliveira de Amorim, Alian Aguila, Liliane Rosa Alves Manaças.

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
