## [Decision Letter · Decision Letter 0]

24 Nov 2022

PONE-D-22-23301Clinical and pharmacological factors associated with mortality in patients with COVID-19 in a high complexity hospital in Manaus: a retrospective studyPLOS ONE

Dear Dr. Badin,

Thank you for submitting your manuscript to PLOS ONE. After careful consideration, we feel that it has merit but does not fully meet PLOS ONE’s publication criteria as it currently stands. Therefore, we invite you to submit a revised version of the manuscript that addresses the points raised during the review process.

We look forward to receiving your revised manuscript.

Kind regards,

Benjamin M. Liu, MBBS, PhD, D(ABMM), MB(ASCP)

Academic Editor

PLOS ONE

Journal Requirements:

2. In the Methods section of your manuscript, if you are reporting a retrospective study of medical records or archived samples, please ensure that you have discussed whether all data were fully anonymized before you accessed them and/or whether the IRB or ethics committee waived the requirement for informed consent. If patients provided informed written consent to have data from their medical records used in research, please include this information

Reviewers' comments:

Reviewer's Responses to Questions

**Comments to the Author**

1. Is the manuscript technically sound, and do the data support the conclusions?

Reviewer #1: Partly

2. Has the statistical analysis been performed appropriately and rigorously? 

Reviewer #1: Yes

3. Have the authors made all data underlying the findings in their manuscript fully available?

Reviewer #1: Yes

4. Is the manuscript presented in an intelligible fashion and written in standard English?

Reviewer #1: Yes

5. Review Comments to the Author

Reviewer #1: Badin R. et al reported a retrospective study of clinical and pharmacological factors with mortality in patients with COVID-19 in a high complexity hospital in Manaus, Brazil. They included patients from a reference hospital belonging to the Brazilian public health system, in Manaus, from March 2020 to July 2021 and analyzed the clinical and demographic features, the presence of comorbidities, and pharmacotherapeutic management in patients hospitalized for diagnosis of COVID-19, in addition to identifying predictive factors for mortality. After multivariate logistic regression analysis, age, need for mechanical ventilation, length of hospital stay, and penicillin use were associated with death. The topic is interesting and has important role in clinical application. However, some issues are still needed to addressed:

1. As we all know, different genotypes of SARS-CoV-2 has different ability of infection and pathogenesis. The authors should also analyze the genotypes influence on the mortality.

2. It is important to analyze the laboratory parameters (such as CRP, ALT, AST, pCO2, pO2, etc.), and radiological findings at admission with relationship with mortality.

3. It is also important to analyze the demographic and laboratory parameters to find multivariate predictors of in-hospital mortality of patients with COVID-19.

4. Please check and modify the format of the references one by one.

6. PLOS authors have the option to publish the peer review history of their article (what does this mean?). If published, this will include your full peer review and any attached files.

Reviewer #1: **Yes: **Kuanhui Xiang

---

## [Author Response · Author response to Decision Letter 0]

8 Jan 2023

Dear Dr. Chenete and reviewers,

We are pleased to forward the responses regarding the observations by the reviewers of the manuscript “Clinical and pharmacological factors associated with mortality in patients with COVID-19 in a high complexity hospital in Manaus: a retrospective study”, co-authored by Rebeka Caribé Badin, Robson Luís Oliveira de Amorim, Alian Aguila, Liliane R. A. Manaças, which we submit for publication in Plos one. 

1. As we all know, different genotypes of SARS-CoV-2 has different ability of infection and pathogenesis. The authors should also analyze the genotypes influence on the mortality.

During the study period, the genotypic characterization of the SARS-COV-2 strains was carried out by sampling, making it impossible to analyze the impact of this variable on the mortality rate. However, data from the literature (ref: 23-25) and from the epidemiological bulletin of the city of Manaus (ref: 7) report the predominance of the B.1.195 variant in the first wave of infection (June and November 2020) and the P.1 variant in the second wave (December 2020 and February 2021). This information is described in the article discussion.

2. It is important to analyze the laboratory parameters (such as CRP, ALT, AST, pCO2, pO2, etc.), and radiological findings at admission with relationship with mortality.

Recognizing that laboratory and image parameters are also extremely relevant to guide the clinical decision and considering the reviewer's suggestion, we included in the study the analysis of the AST, ALT, and oxygen saturation variables. However, due to the lack of data in the patient’s medical records, it was not possible to include the other suggested variables in the study. The analysis of the collected parameters was incorporated into the reviewed version of the manuscript.

3. It is also important to analyze the demographic and laboratory parameters to find multivariate predictors of in-hospital mortality of patients with COVID-19. 

 All of parameters included in the research (demographic, clinical, laboratory, and pharmacological) were statistically analyzed. For quantitative variables, the Kolmogorov-Smirnov Normality test was applied. Student's t-test (Normal Distribution) and Mann-Whitney (Non-Normal Distribution) were used to compare two groups (survivors and non-survivors). Categorical variables were shown in percentages or absolute values , and the Chi-Square Test and Fisher's Exact Test were used to verify the association between categorical variables. The statistically significant variables identified in the univariate analysis were included in the multivariate logistic regression analysis (model 1) (S1). 

However, some variables did not prove to be statistically significant in model one, such as the need for ICU, the use of corticosteroids, antimicrobials, anticoagulants, antifungals, and an antibiotic class of glycopeptides.

Thus, a second multivariate logistic regression analysis model was done using the reverse stepwise approach until all variables were significant (model 2) (table 3). In the article, we rewrote this information in the topic “characteristics associated with mortality” to clarify the multivariate logistic regression analysis.

4. Please check and modify the format of the references one by one.

We checked and corrected the references, according to the journal's instructions. For this purpose, we use the Zotero program.

---

## [Decision Letter · Decision Letter 1]

11 Jan 2023

Clinical and pharmacological factors associated with mortality in patients with COVID-19 in a high complexity hospital in Manaus: a retrospective study

PONE-D-22-23301R1

Dear Dr. Badin,

We’re pleased to inform you that your manuscript has been judged scientifically suitable for publication and will be formally accepted for publication once it meets all outstanding technical requirements.

Kind regards,

Benjamin M. Liu, MBBS, PhD, D(ABMM), MB(ASCP)

Academic Editor

PLOS ONE

Additional Editor Comments (optional):

Reviewers' comments:

Reviewer's Responses to Questions

**Comments to the Author**

1. If the authors have adequately addressed your comments raised in a previous round of review and you feel that this manuscript is now acceptable for publication, you may indicate that here to bypass the “Comments to the Author” section, enter your conflict of interest statement in the “Confidential to Editor” section, and submit your "Accept" recommendation.

Reviewer #1: All comments have been addressed

2. Is the manuscript technically sound, and do the data support the conclusions?

Reviewer #1: Yes

3. Has the statistical analysis been performed appropriately and rigorously? 

Reviewer #1: Yes

4. Have the authors made all data underlying the findings in their manuscript fully available?

Reviewer #1: Yes

5. Is the manuscript presented in an intelligible fashion and written in standard English?

Reviewer #1: Yes

6. Review Comments to the Author

Reviewer #1: The answers provided by the authors are satisfy for me. they also re-analyzed the data and made it more accurate. I have no any other questions.

7. PLOS authors have the option to publish the peer review history of their article (what does this mean?). If published, this will include your full peer review and any attached files.

Reviewer #1: **Yes: **no

---

## [Editor Report · Acceptance letter]

16 Jan 2023

PONE-D-22-23301R1 

Clinical and pharmacological factors associated with mortality in patients with COVID-19 in a high complexity hospital in Manaus: a retrospective study 

Dear Dr. Badin:

I'm pleased to inform you that your manuscript has been deemed suitable for publication in PLOS ONE. Congratulations! Your manuscript is now with our production department. 

Kind regards, 

on behalf of

Dr. Benjamin M. Liu 

Academic Editor

PLOS ONE